# Two-Dimensional ZnS/SnS_2_ Heterojunction as a Direct Z-Scheme Photocatalyst for Overall Water Splitting: A DFT Study

**DOI:** 10.3390/ma15113786

**Published:** 2022-05-26

**Authors:** Xing Chen, Cuihua Zhao, Hao Wu, Yong Shi, Cuiting Chen, Xi Zhou

**Affiliations:** 1School of Resources, Environment and Materials, School of Chemistry & Chemical Engineering, Guangxi University, Nanning 530004, China; chenxing1915301004@163.com (X.C.); w872240906@163.com (H.W.); shiyong19970616@163.com (Y.S.); 18269002412@163.com (C.C.); 1914402033@st.gxu.edu.cn (X.Z.); 2Guangxi Key Laboratory of Processing for Non-Ferrous Metal and Featured Materials, Guangxi University, Nanning 530004, China

**Keywords:** ZnS/SnS_2_ heterojunction, direct Z-scheme, photocatalysis, water splitting, DFT

## Abstract

Direct Z-scheme photocatalysts have attracted extensive attention due to their strong redox ability and efficient separation of photogenerated electron-hole pairs. In this study, we constructed two types of ZnS/SnS_2_ heterojunctions with different stacking models of ZnS and SnS_2_ layers, and investigated their structures, stabilities, and electronic and optical properties. Both types of heterojunctions are stable and are direct Z-scheme photocatalysts with band gaps of 1.87 eV and 1.79 eV, respectively. Furthermore, their oxidation and reduction potentials straddle the redox potentials of water, which makes them suitable as photocatalysts for water splitting. The built-in electric field at the heterojunction interface improves the separation of photogenerated electron-hole pairs, thus enhancing their photocatalytic efficiency. In addition, ZnS/SnS_2_ heterojunctions have higher carrier mobilities and light absorption intensities than ZnS and SnS_2_ monolayers. Therefore, the ZnS/SnS_2_ heterojunction has a broad application prospect as a direct Z-scheme visible-light-driven photocatalyst for overall water splitting.

## 1. Introduction

As a clean and renewable energy source, hydrogen has attracted significant attention for addressing the increasingly severe environmental problems and fossil energy shortage [1,2,3,4]. The traditional synthesis method of hydrogen relies on electrolysis; however, it will cause other pollution during the electron power generation process. Photocatalytic hydrogen production was first proposed by Fujishima [5]. It only consumes solar energy and produces no pollution, and has been extensively studied [6,7,8,9,10]. In many cases, most of the materials used in this method are wide bandgap semiconductors and only absorb ultraviolet light, resulting in low utilization of solar energy [8,11]. Furthermore, the recombination of photogenerated electrons and holes also leads to the reduction of catalytic efficiency.

Compared with the single semiconductor, the formation of heterojunction has been proved to be one of the most promising ways for the preparation of advanced photocatalyst because of its feasibility and effectiveness for the spatial separation of electron-hole pairs [12,13,14,15,16,17]. Basically, there are three types of conventional heterojunctions: straddling bandgap type (type-I), staggered bandgap type (type-II), and broken bandgap type (type-III) [15], as shown in Figure 1. For type-I heterojunction, the electrons and holes will accumulate on the conduction band (CB) and valence band (VB) of semiconductor B with lower redox potential. Electron-hole pairs cannot be effectively separated because both electrons and holes are accumulated on the same semiconductor. Furthermore, the separation of electron-hole pairs cannot occur in type-III heterojunction because the bandgaps of two semiconductors do not overlap [16]. In type-II heterojunction, electrons and holes respectively accumulate on the CB of semiconductor B with lower reduction potential and the VB of semiconductor A with lower oxidation potential. Although electrons and holes are separated, the strong oxidation and reduction abilities are sacrificed. To solve the problem, direct Z-scheme heterojunction was designed [15]. Direct Z-scheme heterojunction and type-II heterojunction have similar band structures, but their charge-carrier migration mechanism is different. The direct Z-scheme system has a charge-carrier migration pathway that resembles the letter “Z”. Electrons and holes respectively accumulate on the CB of semiconductor A and VB of semiconductor B, preserving strong redox ability and spatially separated sites for reduction and oxidation [17]. As a result, the direct Z-scheme heterojunction has become a research hotspot in the field of photocatalysis. It is important to note that a direct Z-scheme heterojunction can be constructed only if the two semiconductors have the proper band alignment. 

As we know, wurtzite zinc sulfide (ZnS) possesses a wide bandgap of 3.77 eV [18]. The negative potential of free electrons and the highly positive potential of holes in ZnS would ensure that the separated intrinsic carriers can participate in hydrogen synthesis from the redox reactions of pure H_2_O [6]. In addition, it was found that ZnS has an advantage for the rapid generation of electron-hole pairs in photocatalysis [7,8]. Therefore, as a low-cost, non-toxic and naturally abundant semiconductor, ZnS is considered a promising catalytic material. Unfortunately, the large band gap makes it a UV-responsive and visible-blind semiconductor, which results in low utilization of solar energy [18]. In addition, the inherent defect of a single semiconductor—the rapid recombination of photogenrated carriers—also limits its photocatalytic efficiency. Many strategies, such as element doping [11,19,20], defect engineering [21,22,23], and decorating with noble metals [24,25], have been attempted to improve the visible light response of ZnS. Nevertheless, these methods are costly or not easily controllable. Numerous studies show that ZnS-based heterostructure has better photocatalytic performance [9,10,12,13,14].

The semiconductor SnS_2_ possesses the advantages of favorable stability, low toxicity, and natural abundance, which, with a suitable band gap (2.1 eV–2.43 eV) [26,27,28] has been reported for photocatalytic hydrogen production [29,30] and organic degradation [31,32]. For instance, Yu et al. [30] prepared two-dimensional SnS_2_ nanosheets with a thickness of ca. 22 nm and found that they exhibit high photocatalytic H_2_ evolution activity of 1.06 mmol h^−1^ g^−1^ under simulated sunlight irradiation. Nevertheless, the fast recombination of electron-hole pairs also restricts its application. SnS_2_ is a hexagonal semiconductor with van der Waals force connection in the (001) direction, which makes it easy to obtain ultra-thin SnS_2_ nanosheets [33]. Some heterojunctions based on SnS_2_ have been reported [34,35,36,37].

ZnS/SnS_2_ heterojunctions have also been reported [38,39,40,41,42,43], as shown in Table 1. However, different research groups obtained different results. Some researchers [38,39] believed that ZnS/SnS_2_ forms a type-Ⅰ heterojunction. The CB and VB of ZnS are respectively higher and lower than the corresponding bands of SnS_2_, and electrons and holes migrate to the CB and VB of SnS_2_. Other researchers [40,41] believed that ZnS/SnS_2_ forms a type-II heterojunction. The CB and VB of ZnS are higher than the corresponding bands of SnS_2_. In a heterojunction, photogenerated electrons migrate from the CB of ZnS to the CB of SnS_2_, and photogenerated holes transfer from the VB of SnS_2_ to the VB of ZnS, which realizes the separation of photogenerated electron-hole pairs. Still others though that ZnS/SnS_2_ belongs to the direct Z-scheme heterojunction, and their descriptions for energy band alignments of ZnS and SnS_2_ are controversial [42,43]. Derikvandi et al. [42] believed that both the CB and VB of SnS_2_ are lower than the corresponding bands of ZnS, while Xia et al. [43] believed that both the CB and VB of SnS_2_ are higher than the corresponding bands of ZnS. Hence the interlayer charge transfer in the heterojunction is opposite in their studies, and the sites of redox reaction are also different. As a result, the transfer of photogenerated carriers in a ZnS/SnS_2_ heterojunction and the band alignments still need to be further studied.

In our paper, the ZnS/SnS_2_ heterojunction was constructed by the ZnS layer vertically stacking on top of the SnS_2_ layer, and the electron properties, band alignment, and photocatalytic mechanism of heterojunction were further studied by density functional theory (DFT). The charge transfer of the ZnS/SnS_2_ heterojunction was analyzed by the charge density difference and Bader charge. To revel the photocatalytic performance of heterojunction, the carrier mobilities and optical properties of heterojunction are calculated. These results are critical for understanding the photocatalytic mechanism of 2D ZnS/SnS_2_ heterojunction.

## 2. Computational Methods

All calculations in this paper were carried out by the first-principle method based on density functional theory (DFT), as performed in the ab initio VASP code [44,45]. The generalized gradient approximation (GGA) and Perdew-Burke-Ernzerhofer (PBE) functional were used to deal with the electron exchange-correlation functional [46]. The projected augmented wave (PAW) method was adopted to interpret the electron-ion interactions [47,48]. The weak interactions in the layered materials were described using the DFT-D3 method [49]. A plane wave cutoff energy was set to be 500 eV, with a convergence of energy and force of 10^−5^ eV/Å and 0.01 eV/Å. The first Brillouin zone was sampled with a Monkhorst-Pack grid of 6 × 6 × 1. Meanwhile, a vacuum distance of 20 Å along the Z-direction was adopted to avoid periodic boundaries. Because the PBE functional and GGA approaches usually underestimate the band gaps of semiconductors, the screened hybrid Heyd-Scuseria-Ernzerhof (HSE06) functional [50] was employed to obtain accurate electronic structures and optical properties. Ab initio molecular dynamics (AIMD) simulations were used to investigate the thermal stabilities of the ZnS/SnS_2_ van der Waals heterostructures. AIMD simulations were adopted at 300 K, with a total simulation time of 5 ps in 1 fs time steps, which is controlled by the Nose-thermostat algorithm [51].

## 3. Results and Discussion

### 3.1. Geometric Structures and Stability of Heterojunction

The monolayers of ZnS and SnS_2_ were obtained by cutting hexagonal ZnS and SnS_2_ along the (001) direction, and those are shown in Figure 2a,d, respectively. It is found that the ZnS monolayer undergoes an obvious structural change after optimization (Figure 2b). Its lattice constant is a = b = 3.92 Å, which is larger than that before optimization (a = b = 3.82 Å). Nevertheless, there is no obvious change after optimization for the structure of the SnS_2_ layer (Figure 2e) with its lattice constant of a = b = 3.70 Å (before optimization: a = b = 3.64 Å), which is consistent with the previous report [52]. There is a small lattice mismatch rate (5.7%) between ZnS and SnS_2_ due to their similar lattice constants, which makes them easy to form a stable heterostructure in the two-dimensional direction. For the ZnS monolayer, there are two types of Zn-S bonds with a band length of 2.59 Å in the vertical direction and 2.33 Å in the other direction (Figure 2b). There is only one type of Sn-S bond for the SnS_2_ monolayer, and its bond length is 2.60 Å. The electronic band structures of the ZnS and SnS_2_ monolayer were calculated with the hybrid HSE06 functional, respectively, as shown in Figure 2c,f. It is observed that the ZnS monolayer is a direct band gap semiconductor with a band gap of 3.87 eV, while the SnS_2_ monolayer is an indirect band gap semiconductor with a band gap of 2.38 eV. All these results are in good agreement with previous reports [9,15,53].

To search for the most stable configuration of the ZnS/SnS_2_ heterojunction, six possible stacking models, namely, H1, H2, H3, H4, H5, and H6 are constructed (Figure 3). Among them, H1 and H4 are the original models, H2 and H3 models are obtained by the horizontal movement of the ZnS layer based on the H1 model, and H5 and H6 models are obtained by the horizontal movement of the ZnS layer based on the H4 model. The energy and force convergences of the six configurations were carried out, and the details of the structure and electronic properties of the different configurations are listed in Table 2. Because the heterojunction still maintains the symmetry of lattice and low lattice mismatch rate, no obvious structural change is observed for the ZnS/SnS_2_ heterojunction, namely the bond lengths of Zn-S and Sn-S did not change significantly.

To quantitatively study the thermodynamic stability of the ZnS/SnS_2_ heterojunction, we calculated the binding energy of heterojunction using the PBE functional. The binding energy ($E_{b}$) is evaluated by the following equation:(1)$$E_{b}=\left(E_{ZnS/SnS_{2}}-E_{ZnS}-E_{SnS_{2}}\right)/A$$
where $E_{ZnS/SnS_{2}}$, $E_{ZnS}$, and $E_{SnS_{2}}$ imply the total energies of the ZnS/SnS_2_ heterojunction, ZnS monolayer, and SnS_2_ monolayer, respectively, and A is the interface area. If the binding energy is negative, it indicates that the model of the heterojunction is stable, and the more negative the value is, the more stable the structure is. It can be seen from Table 1 that the binding energies of all models are negative; however, the binding energies of H2, H3, H4, and H6 are similar, which are more negative than those of H1 and H5, showing the models of H2, H3, H4, and H6 are more stable than those of H1 and H5. The values of the binding energies for H2, H3, H4, and H6 are close to the results of a typical heterojunction, such as transition-metal dichalcogenide/BSe (−17.47 meV/Å^2^) [54] and GaTe/CdS (−13.56 meV/Å^2^) [55]. In the four models (H2, H3, H4, and H6), the binding energy of H4 is the most negative, −18.63 meV/Å^2^, then H3, −18.58 meV/Å^2^, H2, −18.37 meV/Å^2^; and H6, −18.19 meV/Å^2^. The H3 and H4 models with the most negative binding energies will be discussed in the following sections.

AIMD simulations were adopted to further explore the thermal stability of the H3 and H4 heterojunctions with a supercell of 4 × 4 × 1, and the temperature was set to 300 K. Figure 4a,b are the fluctuation of the total energy of H3 and H4 heterojunctions, respectively. It is observed that H3 and H4 heterojunctions have no obvious structural deformation, and there is no bond cleavage after running for 5000 fs. In addition, it is clear that the fluctuations of total energy are relatively large only at the beginning of operation for the H3 and H4 heterojunctions, and then the fluctuations become small with extending time to 5000 fs. The average value of total energy remains almost unchanged between 1000 fs and 5000 fs. These results suggest the stability of H3 and H4 heterojunctions at room temperature [56,57]. We also note that the fluctuation of total energy for the H4 heterojunction is smaller than that for the H3 heterojunction, showing that the H4 heterojunction is more stable than the H3 heterojunction, which is well in agreement with the result of the binding energies (Table 2). The binding energy of the H4 heterojunction is more negative than that of the H3 heterojunction.

### 3.2. Electronic Properties

In order to investigate the electronic properties of the heterojunction, we calculated the projected band structure [58] of the H3 and H4 heterojunction by HSE06 hybrid functional, as shown in Figure 5a,d. The H3 and H4 heterojunctions are direct band gap semiconductors with values of 1.87 eV and 1.79 eV, respectively. Their CBMs and VBMs are situated at the high-symmetric Γ point of the Brillouin zone. Compared with the band gaps of the ZnS layer (3.87 eV) and the SnS_2_ layer (2.38 eV), the band gaps of the H3 and H4 heterojunctions are much smaller, which is attributed to the staggered alignment of the energy bands of ZnS and SnS_2_. It is also noted that the CBM and VBM of the H3 and H4 heterojunctions are from the SnS_2_ and ZnS layers, respectively.

Figure 5b,e show the partial density of states (PDOS) of the H3 and H4 heterojunctions. It can be seen that the CBMs of the H3 and H4 heterojunctions are occupied by S and Sn atomic orbitals of SnS_2_, and the VBMs are occupied by S and Zn atomic orbitals of ZnS. To give a more vivid representation, the local charge density of the Highest-Occupied-Molecular-Orbital (HOMO) and the Lowest-Unoccupied-Molecular-Orbital (LUMO) were calculated, as depicted in Figure 5c,f. The HOMO is almost concentrated upon the ZnS_2_ layer, while the LUMO is mainly located on the SnS_2_ layer, suggesting that the ZnS layer easily loses electrons, while the SnS_2_ layer easily obtains electrons, which is consistent with the results of energy band and corresponding PDOS. The energy levels from the atomic orbitals of the ZnS layer are distributed at the top of the valence band of the H3 and H4 heterojunctions, while those from the atomic orbitals of the SnS_2_ layer are distributed at the bottom of the conduction band (Figure 5a,b,d,e). These results also indicate that the H3 and H4 heterojunctions form the staggered band alignment structures, which can facilitate the effective separation of photogenerated carriers in H3 and H4 heterojunctions.

In addition, the three-dimension charge density difference (CDD) was calculated to explore the charge transfer and separation at the interface of the H3 and H4 heterojunctions, as depicted in Figure 6a,d. The charge accumulation is shown as the yellow region, and the charge depletion is shown as the cyan region. The CDD was calculated by the following equation:(2)$$\Delta \rho \left(z\right)=\rho_{Zn/SnS_{2}}-\rho_{ZnS}-\rho_{SnS_{2}}$$
where $\rho_{Zn/SnS_{2}}$, $\rho_{ZnS}$, and $\rho_{SnS_{2}}$ represent the charge density of the ZnS/SnS_2_ heterojunctions isolated ZnS and SnS_2_ monolayers, respectively. By analyzing Figure 6a,d, it is seen that the ZnS layer donates electrons to the SnS_2_ layer, resulting in the formation of p-doping type in the ZnS layer and n-doping type in the SnS_2_ layer. Bader charge [59] analysis shows that about 0.033 electrons for H3 and 0.028 electrons for H4 are transferred from the ZnS layer to the SnS_2_ layer at the interface of heterojunction. These results are similar to the electrons transfer of the g-GaN/MoSSe heterojunction (0.017 e), g-GaN/WSSe heterojunction (0.023 e) [60].

In heterojunctions, the gradient of plane-average electrostatic potential at the interface affects electrons transfer [54,61]. Figure 6b,e show the average electrostatic potential for the H3 and H4 heterojunctions. It is obvious that the SnS_2_ layer has a lower potential than the ZnS layer, leading to the migration of electrons from the ZnS layer to the SnS_2_ layer, which is confirmed by the Planar-averaged electron density difference, as illustrated in Figure 6c,f. Planar-averaged electron density difference was calculated by integrating inplane CDD, which is given by the following:(3)$$\Delta \rho \left(z\right)=\int \rho_{ZnS/SnS_{2}}\left(x,y,z\right)dxdy-\int \rho_{ZnS}\left(x,y,z\right)dxdy-\int \rho_{SnS_{2}}\left(x,y,z\right)dxdy$$
where $\int \rho_{ZnS/SnS_{2}}\left(x,y,z\right)$ is charge density at point (x,y,z) of the ZnS/SnS_2_ heterojunction, and $\int \rho_{ZnS}\left(x,y,z\right)$ and $\int \rho_{SnS_{2}}\left(x,y,z\right)$ represent the charge density at point (x,y,z) of the ZnS layer and the SnS_2_ layer, respectively. The transfer of electrons from the ZnS layer to the SnS_2_ layer leads to the formation of a built-in electric field with the direction from ZnS to SnS_2_ at the heterojunction interface, which is beneficial to the separation of photogenerated electrons and holes. The same situation also appeared in MoSSe/BSe and BY/MX_2_ heterojunctions [61,62].

### 3.3. Photocatalytic Water Splitting

The basic requirement for the hydrogen evolution reduction (HER) and oxygen evolution reduction (OER) is the appropriate band edge position. Normally, the CBM value should be higher than the energy level of the reduction potential ($E_{H^{+}/H_{2}}$), and its VBM value should be lower than the energy level of the oxidation potential ($E_{O_{2}/H_{2}O}$) [37,61]. We calculated the relative position of the band edges of the isolated ZnS and SnS_2_ monolayers and heterojunctions with reference to the vacuum level, as shown in Figure 7a. The reduction and oxidation potentials of water are also represented in the figure, and they are are −4.44 eV and −5.67 eV at pH = 0, respectively. It is found that both $E_{H^{+}/H_{2}}$ and $E_{O_{2}/H_{2}O}$ lie inside the band gaps of the isolated SnS_2_ and ZnS, and H3 and H4 heterojunctions, indicating that all these systems can be utilized in water splitting in an acid solution. The SnS_2_ layer and the ZnS layer have a staggered band alignment structure in the heterojunction. The band edge positions of the ZnS layer are different from the previous reports [9,39]. The difference is mainly caused by the band edge position of the ZnS layer. The VBM and CBM calculated by Hao [9] are −7.23 eV vs. Vacuum (2.79 v vs. NHE) and −3.61 eV vs. Vacuum (−0.83 v vs. NHE), and VBM and CBM calculated by Xu [39] are −7.05 eV vs. Vacuum (2.61 v vs. NHE) and −3.35 eV vs. Vacuum (−1.09 v vs. NHE), respectively. We found that the 2D ZnS layer and bulk ZnS have a difference in band edge positions, while that of 2D SnS_2_ and bulk SnS_2_ has little difference. Therefore, the band edge positions of bulk ZnS were calculated, as shown in Appendix A. Compared with 2D ZnS, bulk ZnS has lower CBM (−6.94 eV) and VBM (−3.32 eV) values. The values of CBM and VBM are comparable to those of Xu [39].

Due to the similar band edge positions of the H3 and H4 heterojunctions, only the photocatalytic mechanism diagram of the H4 heterojunction is shown in Figure 7b. When exposed to visible light, the electrons absorb energy from photons and migrate from VBM to CBM. For the ZnS/SnS_2_ heterojunction, the photogenerated electrons in the CBM of SnS_2_ and photogenerated holes in the VBM of ZnS are likely to recombine. As mentioned above, the built-in electric field directed from ZnS to SnS_2_ accelerates their recombination. In addition, the existence of a built-in electric field also prevents the photoexcited electrons in the CBM of ZnS from flowing to the CBM of SnS_2_, and the probability of the photoexcited holes in the VBM of SnS_2_ flowing to the VBM of ZnS is also significantly reduced. Therefore, the photoexcited electrons and holes accumulate in the CBM of ZnS and the VBM of SnS_2_, as shown in Figure 7b. The spatial separation of photoexcited carriers prolongs their lifetime, and effectively improves the photocatalytic efficiency of the ZnS/SnS_2_ heterojunction. The above results indicate that the ZnS/SnS_2_ heterojunction is a direct Z-type photocatalyst.

### 3.4. Carrier Mobility

Recent studies have shown that carrier mobility is a key factor affecting photocatalysis because high carrier mobility ensures the full utilization of redox reactions prior to recombination. Therefore, we adopted the deformation potential (DP) theory to study the carrier mobility of the ZnS/SnS_2_ heterojunction, as shown below [63],
(4)$$\mu =\frac{e\hbar^{3}C_{2D}}{K_{B}Tm^{*}m_{d}E_{1}^{2}}$$
where $C_{2D}$ is the elastic modulus of the material along the transport direction, which is obtained by $C_{2D}=\frac{1}{S_{0}}\left(\frac{\partial^{2}E_{total}}{\partial {\left(\Delta l/l_{0}\right)}^{2}}\right)$, and $E_{total}$, Δl, $l_{0}$, and $S_{0}$ are the total energy, deformation quantity, lattice constant, and area of unstrained supercell, respectively. $K_{B}$ is the Boltzmann constant and T is the temperature (300 K). m* is the effective mass for electrons or holes obtained by $m^{*}=\hbar^{2}{\left(\frac{\partial^{2}E\left(k\right)}{\partial k^{2}}\right)}^{-1}$. $m_{d}$ is the average effective mass of the carrier obtained by $m_{d}={\left(m_{x}^{*}m_{y}^{*}\right)}^{1/2}$, where $m_{x}^{*}$ and $m_{y}^{*}$ represent the effective mass of the carrier in the x-direction and y-direction, respectively. $E_{1}$ is the DP constant obtained by $E_{1}=\frac{\Delta E}{\left(\Delta l/l_{0}\right)}$, and ΔE is the energy difference of CBM or VBM under uniaxial tension. The corresponding parameters are shown in Table 3. Both H3 and H4 heterojunctions have smaller electron effective mass and larger hole effective mass. In the H3 and H4 heterojunctions, the elastic constant C_2D_ is almost equal in the armchair and zigzag directions, approximately equal to 135 J/m^−2^, but the carrier mobility of electrons is significantly larger than that of holes. The mobility of electrons varies significantly in different directions. In the armchair direction, the electron mobilities are 3555.04 cm^2^ V^−1^ S^−1^ (H3) and 6362.56 cm^2^ V^−1^ S^−^^1^ (H4), while in the zigzag direction, the electron mobilities are 2044.07 cm^2^ V^−1^ S^−1^ (H3) and 2399.82 cm^2^ V^−1^ S^−^^1^ (H4). It can be found that electron mobilities are anisotropic in the ZnS/SnS_2_ heterojunction. In contrast, the mobility of holes in different directions changes little. In the armchair direction, the hole mobilities are 218.65 cm^2^ V^−1^ S^−1^ (H3) and 255.96 cm^2^ V^−1^ S^−1^ (H4), while in the zigzag direction, the hole mobilities are 175.74 cm^2^ V^−1^ S^−1^ (H3) and 249.84 cm^2^ V^−1^ S^−1^ (H4). Compared with the H3 heterojunction, the H4 heterojunction has higher carrier mobility, which means higher photocatalytic efficiency. The electron mobility of the ZnS/SnS_2_ heterojunction is significantly higher than that of other 2D materials, such as MoS_2_ (200.52 cm^2^ V^−1^ S^−^^1^) and the MoS_2_/BSe heterojunction (300 cm^2^ V^−1^ S^−^^1^) [28,54]. Thus, based on the excellent carrier mobilities in the ZnS/SnS_2_ heterojunction, a high photocatalytic activity can be anticipated.

### 3.5. Optical Properties

In order to obtain efficient photocatalytic devices, it is particularly important to understand their optical properties. Therefore, we calculated the refractive index, extinction coefficient, and optical absorption of ZnS monolayer, SnS_2_ monolayer, and H3 and H4 heterojunctions by PBE functional, respectively. Figure 8 shows the refractive index (n) and extinction coefficient (k) of the ZnS monolayer, SnS_2_ monolayer, and H3 and H4 heterojunctions. The refractive index (n) and extinction coefficient (k) can be expressed as follows:(5)$$n=\sqrt{\left(\sqrt{\varepsilon_{1}^{2}\left(\omega \right)+\varepsilon_{2}^{2}\left(\omega \right)}+\varepsilon_{1}\left(\omega \right)\right)/2}$$
(6)$$k=\sqrt{\left(\sqrt{\varepsilon_{1}^{2}\left(\omega \right)+\varepsilon_{2}^{2}\left(\omega \right)}-\varepsilon_{1}\left(\omega \right)\right)/2}$$
where ω represents the angular frequency of light in vacuum and $\varepsilon_{1}\left(\omega \right)$ and $\varepsilon_{2}\left(\omega \right)$ are the real and imaginary parts of complex dielectric function, respectively. For refractive index (n), it is found that there are 2 and 3 peaks for the ZnS layer and the SnS_2_ layer in the range of 300 to 800 nm. After the combination of ZnS and SnS_2_, there are 1 and 4 peaks for the H4 and H3 heterojunctions in the range of 300 to 800 nm. Compared with the refractive index of the monolayer, the refractive index intensity of the heterojunction is significantly higher in the visible range (390–780 nm). Meanwhile, the extinction coefficient (k) can effectively affect light absorption of the material [64]. As presented in Figure 8b, the extinction coefficient of the ZnS/SnS_2_ heterojunction is evidently larger than that of the ZnS layer and SnS_2_ layer, when the wavelength is in the range of 430 to 800 nm. The H3 heterojunction has four peaks with values of 1.21, 0.70, 0.42, and 0.21, corresponding to 315 nm, 353 nm, 469 nm, and 561 nm, respectively. However, there are no peaks for the H4 heterojunction.

Figure 9 shows the optical absorption of ZnS, SnS_2_, and ZnS/SnS_2_ heterojunction. The optical absorption coefficient (α) is related to the extinction coefficient (k) by:(7)α=2kω/c
where c represents the speed of light. It is seen that the optical absorption is proportional to k. Similar to the performance of the extinction coefficient, the light absorption coefficients of the H3 and H4 heterojunctions are larger than those of ZnS and SnS_2_ in most of visible light range (430–780 nm), and H3 heterojunction also has four peaks with values of 4.84 × 10^5^ cm^−1^, 2.51 × 10^5^ cm^−1^, 1.13 × 10^5^ cm^−1^, and 0.46 × 10^5^ cm^−1^, corresponding to 315 nm, 353 nm, 469 nm, and 561 nm, respectively. In the range of 300 nm to 430 nm, the optical absorption coefficients of the SnS_2_, H3, and H4 heterojunctions are comparable. In general, the formation of the ZnS/SnS_2_ heterojunction can improve the light absorption and thus enhance the photocatalytic efficiency.

## 4. Conclusions

In summary, we designed a 2D ZnS/SnS_2_ heterojunction and used a first-principle method to calculate its electronic and optical properties, carrier mobility, and potential application in water splitting. Both binding energy calculations and AIMD simulations indicate that the SnS_2_ layer and the ZnS layer can form stable ZnS/SnS_2_ heterojunctions. Two kinds of ZnS/SnS_2_ heterojunctions (H3 and H4) were constructed by different stacking modes of ZnS layer and SnS_2_ layer, and they are both direct Z-scheme type photocatalysts with gap values of 1.87 and 1.79 eV, respectively. The ZnS and SnS_2_ layers are used as photocatalysts of HER and OER separately with strong redox ability to split water into hydrogen and oxygen. Charge transfer occurs at the interface of heterojunctions and induces a built-in electric field with the direction from ZnS to SnS_2_. The built-in electric field accelerates the recombination of electrons from the CBM of SnS_2_ and holes from the VBM of ZnS, and hinders the reverse transfer of charge carriers, which can effectively improve the separation efficiency of photoinduced electron-hole pairs. Moreover, it is found that H3 and H4 heterojunctions have relatively high electron mobility and moderate hole mobility, indicating that they are highly active photocatalysts. Compared with the single ZnS and SnS_2_ layers, the H3 and H4 heterojunctions have higher light absorption intensities in most of the visible light range (430–780 nm). Therefore, the ZnS/SnS_2_ heterojunction is a direct Z-scheme photocatalyst with great potential and wide application for water splitting.

## Figures and Tables

**Figure 1 materials-15-03786-f001:**
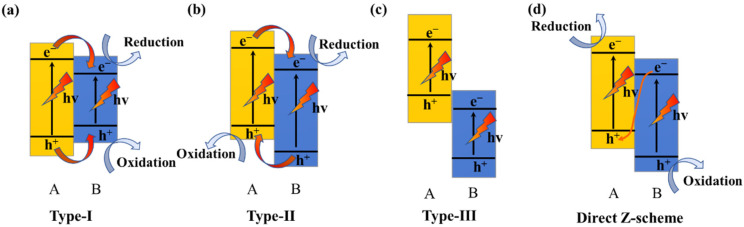
The charge-carrier separation mechanism on the type-I heterojunction (**a**), type-II heterojunction (**b**), type-III heterojunction (**c**), and the direct Z-scheme (**d**), and heterojunctions built on two different semiconductors.

**Figure 2 materials-15-03786-f002:**
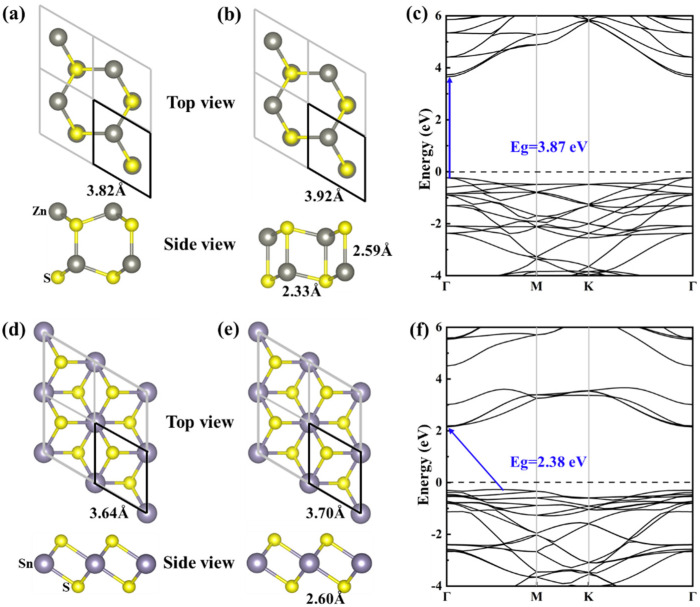
Top and side views of ZnS and SnS_2_ monolayer before geometric optimization (**a**,**d**) and after geometric optimization (**b**,**e**). The sulfur (S), zinc (Zn), and tin (Sn) atoms are indicated by yellow, gray, and violet balls, respectively. The band structures of monolayer ZnS (**c**) and SnS_2_ (**f**).

**Figure 3 materials-15-03786-f003:**
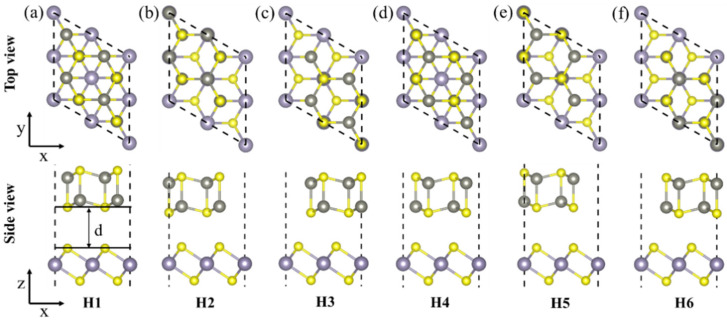
Top and side views of (**a**) H1, (**b**) H2, (**c**) H3, (**d**) H4, (**e**) H5, and (**f**) H6 stacking models of ZnS/SnS_2_ heterojunction after atomic relaxation.

**Figure 4 materials-15-03786-f004:**
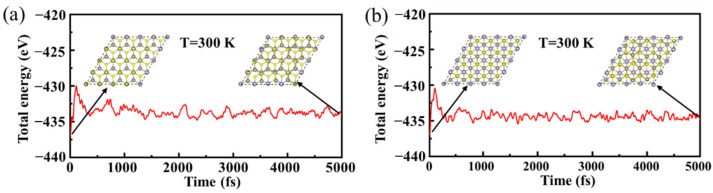
Fluctuation of the total free energy of H3 (**a**) and H4 (**b**) heterojunction during AIMD simulations.

**Figure 5 materials-15-03786-f005:**
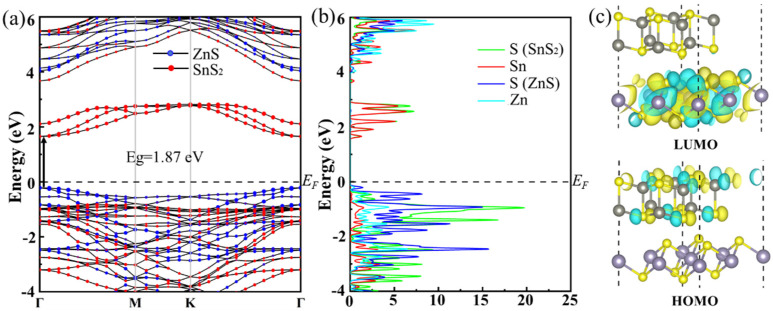
Projected band structure (obtained with the HSE06 function) of H3 (**a**) and H4 (**d**) heterojunction; the blue and red symbols represent the contribution of ZnS and SnS_2_, respectively; the zero energy corresponds to the Fermi level and is represented by the black dashed line. The projected density of states of H3 (**b**) and H4 (**e**) heterojunction. The band decomposed charge density for HOMO (VBM) and LUMO (CBM) of the H3 (**c**) and H4 (**f**) heterojunction, and the value of the isosurface was set to 3 × 10^−8^ e.

**Figure 6 materials-15-03786-f006:**
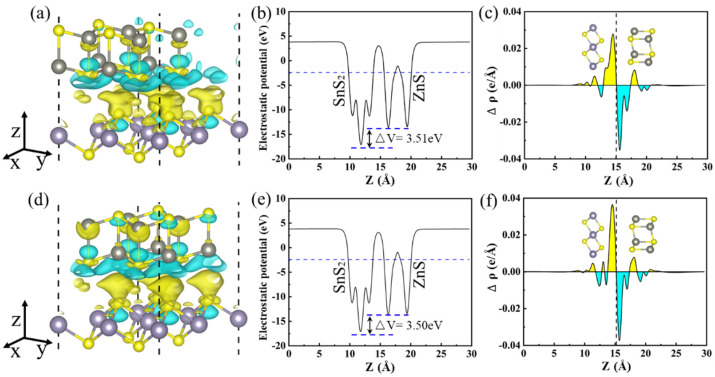
The charge density difference for H3 (**a**) and H4 (**d**) heterojunction and the difference charge density with an isovalue of 0.0001 e Å^−3^. Plane-averaged electrostatic potential drop across the interface of the H3 (**b**) and H4 (**e**) heterojunctions. Planar-averaged electron density difference Δρ(z) for H3 (**c**) and H4 (**f**) heterojunctions. The yellow and cyan areas indicate electron accumulation and depletion, respectively.

**Figure 7 materials-15-03786-f007:**
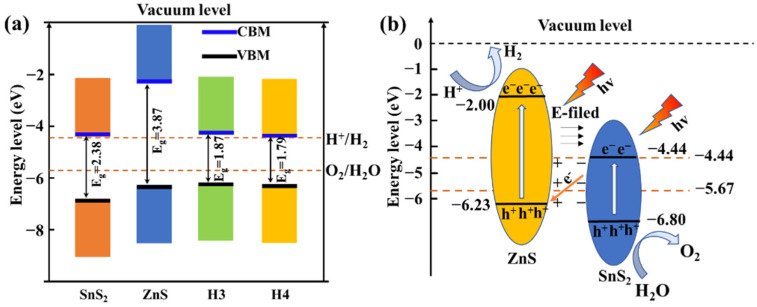
Schematic presentation of the band edges with respect to the vacuum level of the SnS_2_ layer, ZnS layer, and heterojunctions, as well as the reduction (H^+^/H_2_) and oxidation (O_2_/H_2_O) potentials at pH = 0 (**a**). Schematic diagram of photocatalysis of H4 model Z-scheme mechanism (**b**).

**Figure 8 materials-15-03786-f008:**
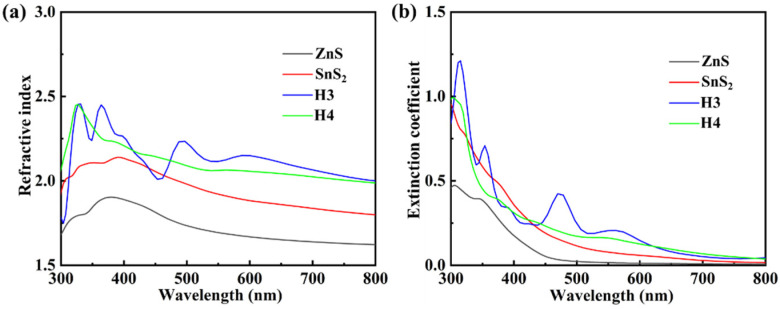
Refractive index (**a**) and extinction coefficient (**b**) for ZnS monolayer, SnS_2_ monolayer, and H3 and H4 heterojunctions.

**Figure 9 materials-15-03786-f009:**
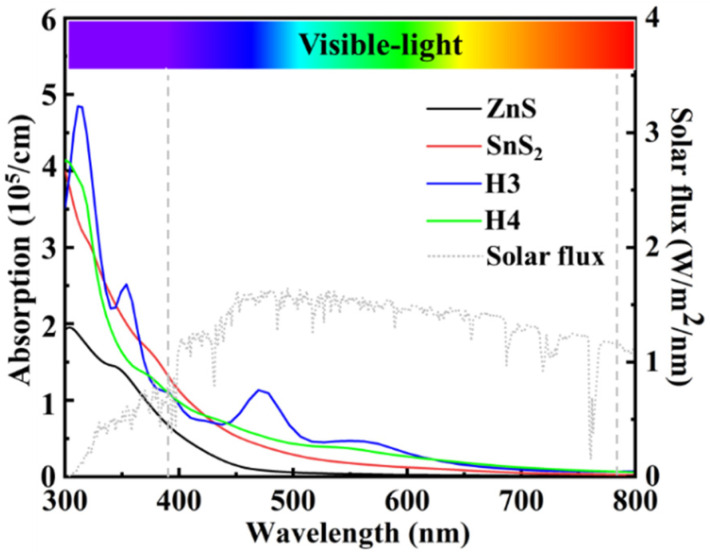
Optical absorption of ZnS monolayer, SnS_2_ monolayer, and ZnS/SnS_2_ (H3 and H4) heterojunctions. The incident AM 1.5G solar flux is shown as a reference.

**Table 1 materials-15-03786-t001:** Some studies on SnS_2_/ZnS heterojunction for various applications.

Sample	Heterojunction Type	Application	Year	Ref.
Pr-SnS_2_/ZnS hierarchical nanoflowers	Type-I	NH_3_ sensing	2019	[38]
SnS_2_/ZnS microspheres	Type- I	Triethylamine detection	2021	[39]
ZnS-SnS_2_ porous nanosheets	Quasi type-Ⅱ	Hydrogen production	2018	[40]
ZnS/SnS_2_ heterojunction	Type-Ⅱ	—	2018	[41]
SnS_2_-ZnS/clinoptilolite	Z-scheme	Photodegradation of phenol	2020	[42]
ZnS/SnS_2_ heterojunction	Z-scheme	Tetracycline degradation	2020	[43]

**Table 2 materials-15-03786-t002:** Calculated equilibrium interlayer distance, d, binding energy, E_b_, other direction, L_Zn-S(a)_, vertical direction, L_Zn-S(b)_, and bond lengths, L_Sn-S_, of ZnS/SnS_2_ heterostructure for six models.

Models	d (Å)	L_Zn-S(a)_ (Å)	L_Zn-S(b)_ (Å)	L_Sn-S_ (Å)	E_b_ (meV/Å^2^)
H1	3.62	2.28	2.61	2.62	−12.03
H2	3.01	2.28	2.62	2.62	−18.37
H3	2.95	2.28	2.62	2.63	−18.58
H4	3.00	2.28	2.62	2.63	−18.63
H5	3.61	2.28	2.61	2.62	−12.05
H6	2.97	2.28	2.61	2.62	−18.19

**Table 3 materials-15-03786-t003:** Values of effective mass (m*), deformation potential constant (E_1_), elastic modulus (C_2D_), and carrier mobility (μ) for H4 heterojunction along the transport directions obtained by the PBE functional; x and y represent armchair and zigzag directions, respectively. The temperature was set at 300 K.

Type	Direction	Carrier Type	m* (m_0_)	E_1_(eV)	C_2D_ (J m^−2^)	μ (cm^2^ V^−1^ S^−1^)
H3	Armchair (x)	e	0.91	−1.30	135.82	3555.04
		h	−1.11	−3.04	135.82	218.65
	Zigzag (y)	e	0.31	−2.93	135.14	2044.07
		h	−1.51	−2.90	135.14	175.74
H4	Armchair (x)	e	0.89	−0.98	133.61	6362.56
		h	−1.43	−2.37	133.61	255.96
	Zigzag (y)	e	0.31	−2.75	138.22	2399.82
		h	−1.35	−2.52	138.22	249.84

## Data Availability

The data presented in this study are available on request from the corresponding author.

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
