# Peer review of "Two-Dimensional ZnS/SnS2 Heterojunction as a Direct Z-Scheme Photocatalyst for Overall Water Splitting: A DFT Study"

_materials, 2022, doi:10.3390/ma15113786_

Round 1
Reviewer 1 Report
The manuscript entitled “Two-Dimensional ZnS/SnS2 Heterojunction as Direct Z-Scheme 2 Photocatalyst for overall water splitting: A DFT study” has been submitted by authors. Some issues to be addressed which will improve the quality of manuscript. Therefore, I recommend this work could be published after the major revision
- The English composition requires many improvements. The authors should proofread the manuscript carefully to minimize grammatical errors.
- The background of this work is not clear. The authors should specify in a clearer way what novel and original this work proposes to readers based on some new works. This research topic is widely studied in past and lot of studies are performed. Author please added comparative table for reader clear understanding.
The characterization part and the result and discussion part are not supported by enough references. It may be supported by the recent relevant references (before 2015).
Reviewer 2 Report
In the manuscript the Authors investigated the structural, energetic, electronic and optical properties of ZnS/SnS2 heterojunction. This is a computational study that is motivated by a potential application of ZnS/SnS2 in the electrochemical photolysis of water. The computational methodology applied by the Authors is correct, the results are presented in a clear manner, the discussion of the results is brief but sufficient for the purpose of the manuscript. In my opinion, the manuscript is free from serious flaws, suits the scope of Materials and it might be interesting to the readers of the journal. Therefore, I can recommend it for publication.
I suggest a couple of minor corrections:
Abstract, line 12: the abbreviations H3 and H4 have no meaning for those who read the abstract only. They should be removed from the abstract.
line 27: "As a clean and renewable energy" -> "As a clean and renewable energy source"
line 38: "but also effectively separate of photogenerated electrons" -- rephrase, please
line 40: "As a result, heterogeneous catalysis has become a research hotspot in the field of photocatalysis" -- rephrase it, please. I think this statement is too general and misleading. This statement should correspond directly to the system of interest and not to heterogeneous catalysis as a whole
line 56: explain briefly what 'g-C3N4' means. Some readers may be unfamiliar with this notation.
line 71: "different people have different research results" -> "different research groups obtained different results"
Introduction: explain briefly the difference between type-I, type-II and Z-scheme heterojunctions. Some readers may be unfamiliar with these notions.
line 310: What do m^*_x and m^*_y mean?
line 380: "its electronic, optical properties" -> "its electronic and optical properties"
line 392: "high active photocatalysts" -> "highly active photocatalysts"
lines 90-93: These findings suit for the section "Conclusions" rather than for the introduction.
line 116: "has" -> "undergoes"
line 130: "agreements" -> "agreement"
line 149: I assume that the E_b energy was calculated using the PBE functional. I would be beneficial for the quality of the study to confirm the ordering in E_b using the HSE06 functional too.
line 190: "function" -> "functional"
Figure 5: in panels (a) and (d) mark the 'z' direction, please
Section 3.2 some equations are inserted into paragraphs, instead of being separated from the text and marked with numbers
Round 2
Reviewer 1 Report
The author carefully addresses all suggested comments. I advised accepting it in its current form.